# A SAMPLING FRAMEWORK FOR VALUE-BASED REINFORCEMENT LEARNING

## ABSTRACT

Value-based algorithms have achieved great successes in solving Reinforcement Learning problems via minimizing the mean squared Bellman error (MSBE). Temporal-difference (TD) algorithms such as Q-learning and SARSA often use stochastic gradient descent based optimization approaches to estimate the value function parameters, but fail to quantify their uncertainties. In our work, under the Kalman filtering paradigm, we establish a novel and scalable sampling framework based on stochastic gradient Markov chain Monte Carlo, which allows us to efficiently generate samples from the posterior distribution of deep neural network parameters. For TD-learning with both linear and nonlinear function approximation, we prove that the proposed algorithm converges to a stationary distribution, which allows us to measure uncertainties of the value function and its parameters.

## 1 INTRODUCTION

Reinforcement learning (RL) targets at learning an optimal policy for sequential decision problems in order to maximize the expected future reward. The value-based algorithms such as Temporal-difference (TD) learning (Sutton, 1988), State–action–reward–state–action (SARSA) (Sutton & Barto, 2018), and Q-learning are frequently used, which play a crucial role for policy improvement. TD-learning aims to estimate the value functions, including state-value function and action-value function, by minimizing the mean-squared Bellman error, where the value functions are often approximated by a function family with unknown parameters. Hence, it is critical to evaluate the accuracy and uncertainty of parameter estimation, which enables uncertainty quantification for the sequential decision at a sequence of states.

In the function approximation TD algorithms such as Deep Q-Network, the parameters are commonly optimized by stochastic gradient descent (SGD) based algorithms. The convergence of these algorithms, including both with linear function approximation (Schoknecht, 2002) and nonlinear function approximation (Fan et al., 2020; Cai et al., 2019), has been extensively studied in the literature. However, SGD suffers from the local trap issue while dealing with nonconvex function approximations such as deep neural networks (DNNs). In order to efficiently and effectively explore the landscape of the complex DNN model, Monte Carlo algorithms such as Stochastic Gradient Langevin Dynamics (SGLD) (Welling & Teh, 2011; Aicher et al., 2019; Kamalaruban et al., 2020) have shown their great potential in escaping from local traps. Moreover, under the Bayesian framework, the Monte Carlo algorithms generate samples from the posterior distribution, which naturally describes the uncertainty of the estimates.

Toward uncertainty quantification for reinforcement learning, it is important to note that the reinforcement learning problem can be generally reformulated as a state-space model. In consequence, the value function parameters can be estimated with Kalman filtering methods such as Kalman Temporal Difference (KTD) (Geist & Pietquin, 2010) and KOVA algorithm (Shashua & Mannor, 2020). Under the normality assumption and for linear function approximation, the Kalman filter approaches are able to provide correct mean and variance of the value function, which enables uncertainty quantification for the sequential decision. However, for nonlinear function approximation, KTD and KOVA algorithms adopt unscented Kalman filter (UKF) (Wan & Van Der Merwe, 2000) and extended Kalman filter (EKF) techniques to approximate the covariance matrices. Both algorithms are computationally inefficient for large scale neural networks. KTD requires $O(p^2)$ for covariance update, where $p$ is the

number of parameters. In each iteration, KOVA calculates a Jacobian matrix that grows linearly with batch size.

In this paper, we have two major contributions: (i) We develop a new Kalman filter-type algorithm for valued-based policy evaluation based on the Langevinized Ensemble Kalman filter (Zhang et al., 2021; Dong et al., 2022).The new algorithm is scalable with respect to the dimension of the parameter space, which has a computational complexity of $O(p)$ for each iteration. (ii) We prove that even when the policy is not fixed, under some regularity conditions, the proposed algorithm converges to a stationary distribution eventually.

## 2 BACKGROUND

### 2.1 MARKOV DECISION PROCESS FRAMEWORK

The standard RL procedure aims to learn an optimal policy from the interaction experiences between an agent and an environment, where the optimal policy maximizes the agent's expected total reward. The RL procedure can be described by a Markov decision process (MDP) represented by $\{\mathcal{S}, \mathcal{A}, \mathcal{P}, r, \gamma\}$, where $\mathcal{S}$ is set of states, $\mathcal{A}$ is a finite set of actions, $\mathcal{P} : \mathcal{S} \times \mathcal{A} \times \mathcal{S} \to \mathbb{R}$ is the state transition probability from state $s$ to state $s'$ by taking action $a$, denoted by $\mathcal{P}(s'|s, a)$, $r(s, a)$ is a random reward received from taking action $a$ at state $s$, and $\gamma \in (0, 1)$ is a discount factor. At each time stage $t$, the agent observes state $s_t \in \mathcal{S}$ and takes action $a_t \in \mathcal{A}$ according to policy $\rho$ with probability $P_\rho(a|s)$, then the environment returns a reward $r_t = r(s_t, a_t)$ and a new state $s_{t+1} \in \mathcal{S}$. For a given policy $\rho$, the performance is measured by the state value function ($V$-function) $V^\rho(s) = \mathbb{E}^\rho[\sum_{t=0}^\infty \gamma^t r_t | s_0 = s]$ and the state-action value function ($Q$-function) $Q^\rho(s, a) = \mathbb{E}^\rho[\sum_{t=0}^\infty \gamma^t r_t | s_0 = s, a_0 = a]$. Both functions satisfy the following Bellman equations:

$$V^\rho(s) = \mathbb{E}^\rho[r(s, a) + \gamma V^\rho(s')],$$
$$Q^\rho(s, a) = \mathbb{E}^\rho[r(s, a) + \gamma Q^\rho(s', a')],$$

where $s' \sim \mathcal{P}(\cdot|s, a)$, $a \sim P_\rho(\cdot|s)$, $a' \sim P_\rho(\cdot|s')$, and the expectations are taken over the transition probability $\mathcal{P}$ for a given policy $\rho$.

### 2.2 BAYESIAN FORMULATION

In this paper, we focus on learning optimal policy $\rho$ via estimating $Q^\rho$. Suppose that Q-functions are parameterized by $Q(\cdot; \theta)$ with parameter $\theta \in \theta \subset \mathbb{R}^p$. Let $\mu_\rho$ be the stationary distribution of the transition tuple $z = (s, a, r, s', a')$ with respect to policy $\rho$. $Q^\rho$ can be estimated by minimizing the mean squared Bellman error (MSBE),

$$\min_\theta \text{MSBE}(\theta) = \min_\theta \mathbb{E}_{z \sim \mu_\rho} \left[ (Q(s, a; \theta) - r - \gamma Q(s', a'; \theta))^2 \right], \tag{1}$$

where the expectation is taken over a fixed stationary distribution $\mu_\rho$. By imposing a prior density function $\pi(\theta)$ on $\theta$, we define a new objective function

$$\begin{aligned} \tilde{\mathcal{F}}(\theta) &= \mathbb{E}_{z \sim \mu_\rho}[\mathcal{F}(\theta, z)] \\ &= \mathbb{E}_{z \sim \mu_\rho} \left[ (Q(s, a; \theta) - r - \gamma Q(s', a'; \theta))^2 - \frac{1}{n} \log \pi(\theta) \right], \end{aligned} \tag{2}$$

where $\mathcal{F}(\theta, z) = (Q(s, a; \theta) - r - \gamma Q(s', a'; \theta))^2 - \frac{1}{n} \log \pi(\theta)$. Since the stationary distribution $\mu_\rho$ is unknown, we consider the empirical objective function

$$\tilde{\mathcal{F}}_{\boldsymbol{z}} = \frac{1}{n} \sum_{i=1}^n \mathcal{F}(\theta, z_i), \tag{3}$$

on a set of transition tuples $\boldsymbol{z} = \{z_i\}_{i=1}^n$. Instead of minimizing $\tilde{\mathcal{F}}_{\boldsymbol{z}}$ directly, one can simulate a sequence of $\theta$ values using the SGLD algorithm by iterating the following equation:

$$\theta_t = \theta_{t-1} - \epsilon_t n F_t(\theta_{t-1}) + \sqrt{2\epsilon_t \beta^{-1}} \omega_t, \tag{4}$$

where $F_t(\theta_{t-1})$ is a conditionally unbiased estimator of $\nabla \mathcal{F}_{\boldsymbol{z}}(\theta_{t-1})$, $\omega_t \sim N(0, I_p)$ is a standard Gaussian random vector of dimension $p$, $\epsilon_t > 0$ is the learning rate at time $t$, and $\beta > 0$ is the constant inverse temperature. It has been proven that under some regularity assumptions, $\theta_t$ converges weakly to the unique Gibbs measure $\pi_{\boldsymbol{z}} \propto \exp(-\beta n \tilde{\mathcal{F}}_{\boldsymbol{z}})$. However, in value-based RL algorithms, the policy $\rho$ is dynamically updated along with parameter $\theta_t$. Therefore, the distribution $\mu_\rho$ of the transition tuple $\boldsymbol{z}$ also evolves from time to time as $\theta_t$ changes. In section 3, we develop a new sampling algorithm, Langivinized Kalman Temporal Difference (LKTD) algorithm, for value-based RL algorithms and establish the convergence of the proposed algorithm under the dynamic policy setting.

## 3 Main Results

In this section, we first introduce the state-space model formulation and the proposed sampling algorithm under the setting of linear function approximation, and then extend the proposed sampling algorithm to the setting of nonlinear function approximation. For simplicity, a full transition tuple with reward and a reduced transition tuple without reward are denoted, respectively, by $z$ and $x$ as

$$z = \begin{cases} (s, a, r, s', a') \\ (s, a, r, s') \end{cases} \quad \text{and} \quad x = \begin{cases} (s, a, s', a') \\ (s, a, s') \end{cases} \quad , \tag{5}$$

for which we often write $z = (r, x)$. The observation function $h(x, \theta)$ is defined as follows:

$$h(x; \theta) = \begin{cases} Q(s, a; \theta) - \gamma Q(s', a'; \theta), \\ Q(s, a; \theta) - \gamma \max_{b \in \mathcal{A}} Q(s', b; \theta), \end{cases} \tag{6}$$

for the SARSA and Q-learning algorithm, where $\gamma$ is the discount factor.

### 3.1 Linear function approximation

Suppose that $\mathcal{Q} = \{Q(\cdot; \theta)\}$ is a family of linear Q-functions, where every Q-function can be approximated in the form

$$Q(s, a; \theta) = \phi(s, a)^\top \theta, \tag{7}$$

where $\phi : \mathcal{S} \times \mathcal{A} \to \mathbb{R}^p$ is a $p$-dimensional vector-valued feature map. For example, $\phi$ can be a polynomial kernel, Gaussian kernel, etc. Let $\boldsymbol{z}_t = (\boldsymbol{r}_t, \boldsymbol{x}_t) = \{(r_{t,i}, x_{t,i})\}_{i=1}^n$ be a batch of transition tuples of size $n$ generated at stage $t$. For convenience, we use bold symbols to represent either vectors or sets of transition tuples depending on the situation. By combining (6) and (7), we define the observation matrix as

$$\Phi(\boldsymbol{x}_t) = \begin{pmatrix} \Phi(x_{t,1}) \\ \vdots \\ \Phi(x_{t,n}) \end{pmatrix}, \tag{8}$$

where each row vector is defined as

$$\Phi(x_{t,i}) = \begin{cases} \left( \phi(s_{t,i}, a_{t,i}) - \gamma \phi(s'_{t,i}, a'_{t,i}) \right)^\top, \\ \left( \phi(s_{t,i}, a_{t,i}) - \gamma \phi(s'_{t,i}, \operatorname{argmax}_{b \in \mathcal{A}} \phi(s'_{t,i}, b)^\top \theta) \right)^\top, \end{cases} \tag{9}$$

for SARSA and Q-learning. Then it is easy to see that the minimization problem of MSBE in (1) can be reformulated as a Bayesian linear inverse problem

$$\boldsymbol{r}_t = \Phi(\boldsymbol{x}_t)\theta + \eta_t, \quad \eta_t \sim N(0, \sigma^2 I), \quad t = 1, 2, \ldots, n, \tag{10}$$

where $\eta_t$ is an additive Gaussian white noise with covariance matrix $\sigma^2 I$, and $\theta$ is subject to the prior distribution $\pi(\theta)$. The corresponding posterior distribution is given by $\pi_*(\theta) \propto e^{-\tilde{\mathcal{F}}(\theta)}$.

To develop an efficient algorithm for simulating samples from the target distribution $\pi_*^\beta(\theta) \propto e^{-\beta \tilde{\mathcal{F}}(\theta)}$, where $\beta$ denotes the inverse temperature, we further reformulate the Bayesian linear inverse model (10) as a state-space model through Langevin diffusion by following Zhang et al. (2021) and Dong et al. (2022):

$$\begin{aligned} \theta_t &= \theta_{t-1} + \frac{\epsilon_t}{2} \nabla \log \pi(\theta_{t-1}) + w_t, \\ \boldsymbol{r}_t &= \Phi(\boldsymbol{x}_t)\theta_t + \eta_t, \end{aligned} \tag{11}$$

where $w_t \sim N(0, \epsilon_t I_p) = N(0, \Omega_t)$, i.e., $\Omega_t = \epsilon_t I_p$, and $\eta_t \sim N(0, \sigma^2 I)$. In the state-space model (11), the state $\theta_t$ evolves in a diffusion process that converges to the prior distribution $\pi(\theta)$. The new formulation does not only allow us to solve the Bayesian inverse problem by subsampling $(\boldsymbol{r}_t, \boldsymbol{x}_t)$ from a given dataset at each stage $t$ (see Theorem S1 of Zhang et al. (2021)), but also allow us to model a dynamic system where the policy $\rho_{\theta_{t-1}}$ changes along with stage $t$. In order to establish the convergence theory of the entire RL algorithm, we impose two fundamental assumptions on the data generating process and the normality structure.

**Assumption 1** *(Data generating process) For each $t$, we are able to generate tuple $z_t \sim \mu_{\theta_{t-1}}$ according to policy $\rho_{\theta_{t-1}}$. Moreover, the stationary distribution $\mu_{\theta_{t-1}}$ has density function $\pi(z_t | \theta_{t-1})$, which is differentiable with respect to $\theta$.*

**Assumption 2** *(Normality structure) For each $t$, let $\boldsymbol{z}_t = \{z_{t,i}\}_{i=1}^n$ be a set of full transition tuples sampled from $\mu_{\theta_{t-1}}$, the conditional distribution $\pi(\boldsymbol{r}_t | \boldsymbol{x}_t, \theta_t)$ is Gaussian:*

$$\boldsymbol{r}_t | \boldsymbol{x}_t, \theta_t \sim N(\Phi(\boldsymbol{x}_t)\theta_t, \sigma^2 I). \tag{12}$$

*That is, $r_{t,i} | x_{t,i}, \theta_t$ are independent Gaussian distributions.*

Figure 1 depicts the RL updating scheme and data generating process. At each stage $t$, the agent interacts with the environment according to the policy $\rho_{\theta_{t-1}}$ and generates a batch of transition tuples $\boldsymbol{z}_t = (\boldsymbol{x}_t, \boldsymbol{r}_t)$ from the stationary distribution $\mu_{\theta_{t-1}}$, which refers to assumption 1. With the artificial normality structure in assumption 2 and the state-space model (11), we combine the RL setting with the forecast-analysis procedure proposed in Zhang et al. (2021) and introduce Algorithm 1. In Theorem 3.1, we

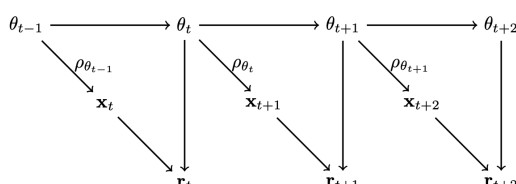

Figure 1: Data generating process

prove that the our algorithm is equivalent to an accelerated preconditioned SGLD algorithm(Li et al., 2016). Then, with some adjustment of Theorem 10 in Raginsky et al. (2017) and the general recipe of stochastic gradient MCMC (Ma et al., 2015), the chain $\{\theta_t^a\}_{t=1}^n$ generated by Algorithm 1 converges to a stationary distribution $p_*(\theta) \propto \exp(-\beta \tilde{\mathcal{G}}(\theta))$ as defined in Lemma A.1 in the Appendix. When $\rho$ is fixed for policy evaluation, Algorithm 1 converges to the target distribution $\pi_*^\beta(\theta)$.

**Algorithm 1** *(Langevinized Kalman temporal difference learning for linear approximation)*

0. *(Initialization) Start with an initial Q-function parameter $\theta_0^a \in \mathbb{R}^p$, drawn from the prior distribution $\pi(\theta)$. For each stage $t = 1, 2, \ldots, T$, do steps 1-3:*

1. *(Sampling) With policy $\rho_{\theta_{t-1}^a}$, generate a set of $n$ transition tuples from the stationary distribution $\mu_{\theta_{t-1}^a}$, denoted by $\boldsymbol{z}_t = (\boldsymbol{r}_t, \boldsymbol{x}_t) = \{z_{t,j}\}_{j=1}^n$, where $z_{t,j}$ has the form of (5). Let $\Phi_t = \Phi(\boldsymbol{x}_t)$.*

   - *Set $\Omega_t = \epsilon_t I_p$, $R_t = 2\sigma^2 I_n$, and the Kalman gain matrix $K_t = \Omega_t \Phi_t^\top (\Phi_t \Omega_t \Phi_t^\top + R_t)^{-1}$.*

2. *(Forecast) Draw $w_t \sim N_p(0, \Omega_t)$ and calculate*

$$\theta_t^f = \theta_{t-1}^a + \frac{\epsilon_t}{2} \nabla \log \pi(\theta_{t-1}^a) + w_t. \tag{13}$$

3. *(Analysis) Draw $v_t \sim N_n(0, R_t)$ and calculate*

$$\theta_t^a = \theta_t^f + K_t(\boldsymbol{r}_t - \Phi_t \theta_t^f - v_t) = \theta_t^f + K_t(\boldsymbol{r}_t - \boldsymbol{r}_t^f). \tag{14}$$

**Theorem 3.1** *Algorithm 1 can be reduced to a preconditioned SGLD algorithm.*

$$\theta_t^a = \theta_{t-1}^a + \frac{\epsilon_t}{2} \Sigma_t \sum_{i=1}^n \nabla \log \pi(\theta_{t-1}^a | z_{t,i}) + e_t, \tag{15}$$

*where $\Sigma_t = (I - K_t \Phi(\boldsymbol{x}_t))$ is a constant matrix given $\boldsymbol{x}_t$, $e_t \sim N(0, \epsilon_t \Sigma_t)$, and $\nabla \log \pi(\theta_{t-1}^a | z_{t,i}) = \frac{1}{\sigma^2} \Phi(x_{t,i})(r_{t,i} - \Phi(x_{t,i})\theta_{t-1}^a) + \frac{1}{n} \nabla \log \pi(\theta_{t-1}^a)$.*

**Theorem 3.2** *Consider the pre-conditioned SGLD algorithm:*

$$\theta_t = \theta_{t-1} - \epsilon_T \Sigma_t G(\theta_{t-1}, \boldsymbol{z}_t) + \sqrt{2\epsilon_T \beta^{-1}} e_t, \quad t = 1, 2, \ldots, T, \tag{16}$$

*where $e_t \sim N(0, \Sigma_t)$, $\beta$ is the inverse temperature, $T$ is the total iteration number, and $\epsilon_T$ is a constant learning rate depending on $T$. For this algorithm, we assume the conditions of Lemma A.1 (of the Appendix) hold. Further, if we choose the learning rate $\epsilon_T$ such that $T\epsilon_T \to \infty$, $T\epsilon_T^{5/4} \to 0$ and $T\epsilon_T \delta^{1/4} \to 0$, then $W_2(p_T, p_*) \to 0$ as $T \to \infty$, where $p_T$ and $p_*$ are as defined in Lemma A.1 of the Appendix.*

**Remark 1** *For the LKTD algorithm, we have $G(\theta_{t-1}, \boldsymbol{z}_t) = -\frac{1}{\sigma^2} \sum_{i=1}^n (\Phi(x_{t,i})^\top (r_{t,i} - \Phi(x_{t,i})\theta_{t-1})) + \frac{1}{\sigma_\theta^2} \theta_{t-1}$. In addition, we set $\epsilon_T = t_0/T^\alpha$ for some constant $t_0 > 0$ and $\alpha \in (4/5, 1)$, such that $T\epsilon_T \delta^{1/4} \to 0$.*

The conditions required by Theorem 3.1 and Theorem 3.2 are verified by Lemma 3.1 given below.

**Lemma 3.1** *Let $G(\theta, \boldsymbol{z}) = -\frac{1}{\sigma^2} \sum_{i=1}^n (\Phi(x_i)^\top (r_i - \Phi(x_i)\theta)) + \frac{1}{\sigma_\theta^2}\theta$. Assume (i) $\mathcal{Z}$ is compact, and $\bar{r} = \sup\{|r| : r \in \mathcal{R}\} < \infty$, (ii) $\|\phi(s, a)\| \leq 1$ for all $s \in \mathcal{S}$, $a \in \mathcal{A}$, and (iii) $\theta_0 \sim N(0, \sigma_0^2 I_p)$, with $\sigma_0^2 < \frac{1}{2}$, then the conditions (A1)-(A6) are satisfied.*

The LKTD algorithm is very flexible. It is not necessary to use all $n$ samples (collected at each time $t$) at each iteration. Instead, a subsample can be used and multiple iterations can be performed for intergrating the available data information in the way of SGLD. Moreover, as explained in Zhang et al. (2021), the forecast-analysis procedure enables the algorithm scalable with respect to the dimension of $\theta_t$, while enjoying the computational acceleration led by the pre-conditioner. In summary, the LKTD algorithm is scalable with respect to both the data sample size and the dimension of the parameter space.

### 3.2 NONLINEAR FUNCTION APPROXIMATION

In this section, we further extend our algorithm to the setting of nonlinear function approximation. For each stage $t$, we consider the nonlinear inverse problem

$$\boldsymbol{r}_t = h(\boldsymbol{x}_t; \theta) + \eta_t, \quad \eta_t \sim N(0, \sigma^2 I), \tag{17}$$

where $h(\boldsymbol{x}; \cdot) : \theta \to \mathbb{R}$ is a nonlinear differentiable observation function of $\theta$. With the state augmentation approach similar to LEnKF algorithm, we define the augmented state vector by

$$\varphi_t = \begin{pmatrix} \theta_t \\ \xi_t \end{pmatrix}, \quad \xi_t = h(\boldsymbol{x}_t; \theta_t) + u_t, \quad u_t = N(0, \alpha\sigma^2 I),$$

where $\xi_t$ is an $n$-dimensional vector, and $0 < \alpha < 1$ is a pre-specified constant. Suppose that $\theta_t$ has a prior distribution $\pi(\theta)$ as we defined in previous section, the joint density function of $\varphi_t = (\theta_t^\top, \xi_t^\top)^\top$ can be written as $\pi(\varphi_t) = \pi(\theta_t)\pi(\xi_t|\theta_t)$, where $\xi_t|\theta_t \sim N(h(\boldsymbol{x}_t; \theta_t), \alpha\sigma^2 I)$. Based on Langevin dynamics, we can reformulate (17) as the following dynamic system

$$\begin{aligned} \varphi_t &= \varphi_{t-1} + \frac{\epsilon_t}{2} \nabla_\varphi \log \pi(\varphi_{t-1}) + w_t, \\ \boldsymbol{r}_t &= H_t \varphi_t + v_t, \end{aligned} \tag{18}$$

where $w_t \sim N(0, \Omega_t)$, $\Omega_t = \epsilon_t I_p$, $p$ is the dimension of $\varphi_t$; $H_t = (0, I)$ such that $H_t \varphi_t = \xi_t$; $v_t \sim N(0, (1-\alpha)\sigma^2 I)$, which is independent of $w_t$ for all $t$. With the formulation in (18), we transformed a nonlinear inverse problem to a linear state-space model and thus the previous theoretical results still hold for the nonlinear inverse problem. The target distribution $p_*(\theta)$ can be easily obtained by marginalization from $p_*(\varphi)$.

**Algorithm 2** *(Langevinized Kalman temporal difference for nonlinear approximation)*

    *0. (Initialization) Start with an initial Q-function parameter ensemble $\theta_0^a \in \mathbb{R}^p$, drawn from the prior distribution $\pi(\theta)$. For each stage $t = 1, 2, \ldots, T$, do the following steps 1-3:*

1. *(Sampling) With policy $\rho_{\theta_{t-1}^a}$ defined in (40), generate a set of $n$ transition tuples from the stationary distribution $\mu_{\theta_{t-1}^a}$ , denoted by $\boldsymbol{z}_t = (\boldsymbol{r}_t, \boldsymbol{x}_t) = \{z_{t,j}\}_{j=1}^n$, where $z_{t,j}$ has the form of (5). Let $H_t = (0, I)$*

    - *For each iteration $k = 1, 2, \ldots, \mathcal{K}$, set $Q_{t,k} = \epsilon_{t,k} I_p$, $R_t = 2(1-\alpha)\sigma^2 I$, and the Kalman gain matrix $K_{t,k} = Q_{t,k} H_t^\top (H_t Q_{t,k} H_t^\top + R_t)^{-1}$, and do steps 2-3.*

2. *(Forecast) Draw $w_{t,k} \sim N_p(0, \Omega_t)$ and calculate*

$$\varphi_{t,k}^f = \varphi_{t,k-1}^a + \frac{\epsilon_{t,k}}{2} \nabla \log \pi(\varphi_{t,k-1}^a) + w_{t,k}, \qquad (19)$$

*where if $k = 1$, set $\varphi_{t,0}^a = (\theta_{t-1,\mathcal{K}}^{a\top}, \boldsymbol{r}_t^\top)^\top$. More precisely, the gradient of two components can be written as*

$$\nabla \log \pi(\varphi_{t,k-1}^a) = \begin{pmatrix} \nabla_\theta \log \pi(\theta_{t,k-1}) + \frac{1}{\alpha\sigma^2} \nabla_\theta h(\boldsymbol{x}_t; \theta_{t,k-1})(\xi_{t,k-1} - h(\boldsymbol{x}_t; \theta_{t,k-1})) \\ -\frac{1}{\alpha\sigma^2}(\xi_{t,k-1} - h(\boldsymbol{x}_t; \theta_{t,k-1})) \end{pmatrix}.$$
$$(20)$$

3. *(Analysis) Draw $v_{t,k} \sim N_n(0, R_t)$ and calculate*

$$\varphi_{t,k}^a = \varphi_{t,k}^f + K_{t,k}(\boldsymbol{r}_t - H_t \varphi_{t,k}^f - v_{t,k}) = \varphi_{t,k}^f + K_{t,k}(\boldsymbol{r}_t - \boldsymbol{r}_{t,k}^f). \qquad (21)$$

## 4 EXPERIMENTS

In this section, we compare LKTD with Adam algorithm (Kingma & Ba, 2014). With a simple indoor escape environment, we show the ability of LKTD in uncertainty quantification and policy exploration. Further, with a more complicated environments such as OpenAI gym, we show that LKTD is able to learn better and more stable policies for both training and testing.

### 4.1 INDOOR ESCAPE ENVIRONMENT

Consider a simple indoor escape environment as shown in Figure 2. The environment is in the square $[0, 1] \times [0, 1]$, and the goal is to reach the top right corner of size $0.1 \times 0.1$ as fast as possible. At the beginning time 0, the agent is randomly put in the square. At each time $t$, the agent observes the location coordinate $s = (x, y)$ as its current state, then chooses an action $a \in \{N, S, E, W\}$ according to policy $\rho$ with a step size randomly drawn from $\text{Unif}([0, 0.3])$. The reward at each time $t$ is $-1$ before the agent reaches the goal. The indoor escaping environment is an example where the optimal policy is not unique. Observe that the Q-values of N and E have no difference in most states, except for the top and the right border. Hence, the ability to explore various optimal policies is critical for learning a stable and robust policy. Through this experiment we show the ability of the LKTD algorithm to learn a mixture optimal policy in a single run. We compare LKTD with the widely used Adam algorithm on training a deep neural network with three

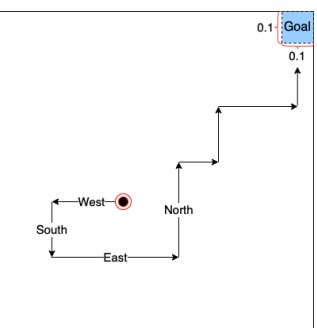

Figure 2: Indoor escape environment

hidden layers of sizes (16,16,16). Agents update the network parameters every 50 interactions and 5 gradient steps per update for a total of 10000 episodes. For action selection, the $\epsilon$-Boltzmann exploration as defined in B.1 is used with an exploring rate of $\epsilon = 0.1$ and an inverse action temperature of $\beta_{\text{act}} = 5$. The batch size is 250. The last 1000 parameter updates are collected as a parameter ensemble, which induces a Q-value ensemble and a policy ensemble. With the Q-value ensemble, we are able to draw the density plot of Q-values at each point of the square as illustrated by Figure 3. To quantify uncertainty of the policy, we define the **mean policy probability** by

$$p_\varrho(a|s) = \frac{1}{|\varrho|} \sum_{\rho \in \varrho} \mathbf{1}_a(\rho(s)), \qquad (22)$$

where $\varrho$ is the policy ensemble induce from the parameter ensemble. Intuitively, the mean policy probability is the proportion of an action taken by the policy ensemble at a given state. We further

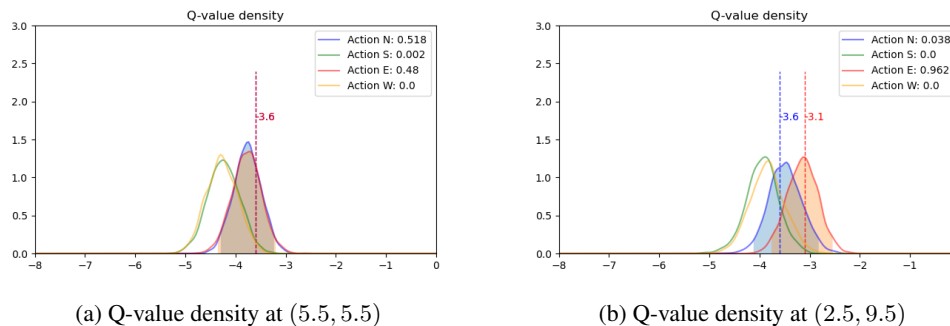

(a) Q-value density at $(5.5, 5.5)$        (b) Q-value density at $(2.5, 9.5)$

Figure 3: Q-value density plots and mean policy probabilities of LKTD

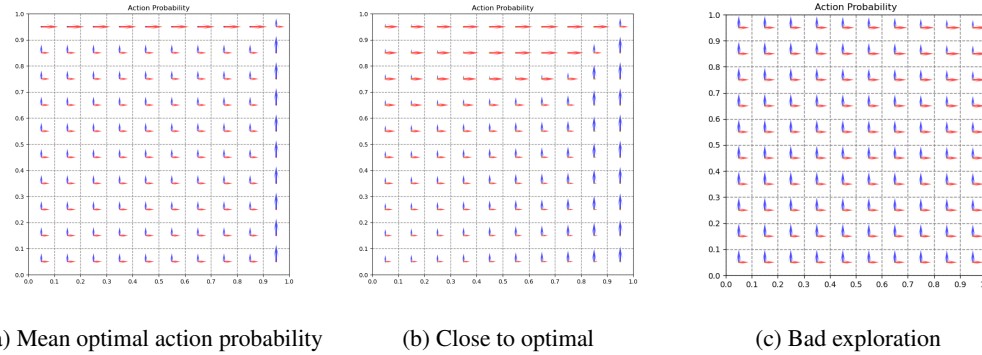

(a) Mean optimal action probability     (b) Close to optimal     (c) Bad exploration

Figure 4: (a) is the mean optimal action probability of indoor escape environment. (b) is similar to (a) on both boundary and interior grids. (c) Fails to find the correct policy on the boundaries, which leads to high false optimal action rate for actions $\{N, E\}$.

define the **mean optimal policy probability** of an environment by taking expectation over all optimal policies. The mean optimal policy probability of the indoor escape environment is shown in figure 4a.

In figure 4, we divide the state space into 100 grids of size $0.1 \times 0.1$, then compute the mean action probabilities of each grid center and each action. For a further comparison of the two algorithms, we calculated two metrics in table 1: (1) MSE between the mean action probability and the mean optimal action probability, denoted by $\text{MSE}(\hat{p})$, where the MSE is taken over all grids. (2) The sub-optimal action rate (SOAR), which is defined by the probability of choosing the action that is sub-optimal for a state. Table 1 shows that in terms of $\text{MSE}(\hat{p})$, LKTD and Adam with large learning rate are more efficient in sample space exploration; however, in terms of SOAR, LKTD is much smaller than Adam in actions $\{N, E\}$, where the high SOAR comes from the top and right boundaries as in figure 4c. In other words, LKTD can efficiently explore the optimal policies, while retaining its accuracy. In figure 3b, the mean policy probability shows that LKTD can choose the correct policy on the boundary grids.

Table 1: $\text{MSE}(\hat{p})$ and SOAR

| Description | | | North | | East | | South | | West | |
|---|---|---|---|---|---|---|---|---|---|---|
| Name | $\epsilon_t$ | $\beta$ | $\text{MSE}(\hat{p})$ | SOAR | $\text{MSE}(\hat{p})$ | SOAR | $\text{MSE}(\hat{p})$ | SOAR | $\text{MSE}(\hat{p})$ | SOAR |
| LKTD | 1e-4 | 1 | 0.044 | 0.098 | 0.044 | 0.102 | 0.002 | 0.026 | 0.002 | 0.026 |
| Adam | 1e-2 | N/A | 0.044 | 0.134 | 0.044 | 0.135 | 0 | 0.005 | 0 | 0.005 |
| Adam | 1e-3 | N/A | 0.047 | 0.493 | 0.047 | 0.495 | 0 | 0 | 0 | 0 |
| Adam | 1e-4 | N/A | 0.058 | 0.502 | 0.058 | 0.495 | 0 | 0 | 0 | 0 |

## 4.2 Classical control problems

In this section, we consider four classical control problems in OpenAI gym (Brockman et al., 2016), including CartPole-v1, MountainCar-v0, LunarLander-v2 and Acrobot-v1. We compare LKTD with Adam under the framework and parameter settings of RL Baselines3 Zoo (Raffin, 2020). Each experiment is duplicated 500 times, and the training progress is recorded in figure 5. At each time step, the best and the worst 1% of the rewards are considered as outliers and thus ignored in the plots. LKTD can also be applied to DQN algorithm by modifying the state-space model in equation 11 as

$$
\theta_t = \theta_{t-1} + \frac{\epsilon_t}{2} \nabla \log \pi(\theta_{t-1}) + w_t,
$$
$$
\boldsymbol{y}_t = \phi(\boldsymbol{s}_t, \boldsymbol{a}_t)^\top \theta_t + \eta_t,
$$
(23)

where $\phi(\boldsymbol{s}_t, \boldsymbol{a}_t) = [\phi(s_{t,1}, a_{t,1}), \dots, \phi(s_{t,n}, a_{t,n})]$ and $\boldsymbol{y}_t = \boldsymbol{r}_t + \gamma \phi(\boldsymbol{s}_t', \boldsymbol{a}_t')^\top \theta_{t-1}$. The new gradient can be written as $G(\theta_{t-1}, \boldsymbol{z}_t) = -\frac{1}{\sigma^2} \sum_{i=1}^n (\phi(x_{t,i})(r_{t,i} - \Phi(x_{t,i})\theta_{t-1})) + \frac{1}{\sigma_\theta^2} \theta_{t-1}$, where the first term corresponds to the semi-gradient in DQN algorithm. With suitable constraints on the semi-gradient, we can modify lemma 3.1 to guarantee the convergence. In the four classic control problems, LKTD shows its strength in efficient exploration and robustness without adopting common RL tricks such as gradient clipping and target network. The updating period of the target network is set to 1 for LKTD. The detail hyperparameter settings are given in section B.4.

In figure 5, the solid and dash lines represent the median and mean rewards, respectively. For each algorithm, the colored area covers 98% of the reward curves. We consider 3 types of reward measurements, training reward, evaluation reward and the best evaluation reward. Training reward records the cumulative reward during training, which include the $\epsilon$-exploration errors. Evaluation reward calculates the mean reward over 10 testing trails at each time $t$. The best evaluation reward only records the best evaluation reward up to time $t$.

In CartPole-v1, LKTD outperforms Adam in all 3 measurements, especially on the training and best evaluation rewards. During training, LKTD receives significantly higher rewards than Adam. In optimal policy exploration, almost 99% of the time LKTD achieves the optimal policy faster than the median of Adam.

In MountainCar-v0, the mean and median reward curves of LKTD and Adam are similar. However, LKTD is more robust during training and more efficient in exploration of good policies. From the best evaluation reward plot, we can observe that the 1% reward lower bound is close to -200 for Adam, which indicates that the agent fails find any good policies during the training.

In order to learn the optimal policies in Lunarlander-v2, the agent has to learn a correct way of landing instead of staying in the air. Due to the sampling nature of LKTD, the exploration rate $\epsilon$ is increased from 0.12 to 0.25 for agent to collect enough landing experiences. Hence, LKTD converges slightly slower than Adam. However, with a large exploration rate, LKTD is still able to obtain stable training rewards which are close to Adam with a much higher lower bound. Moreover, with a longer training period, LKTD will eventually perform better in evaluation.

In Acrobot-v1, the training reward of LKTD converges slower in some experiments, but in most cases, the performance of LKTD dominates Adam.

According to the experiments, LKTD has a more robust training process and finds the optimal polices faster than Adam. The experiments also indicate that Adam uses the rare experiences more efficiently, whereas LKTD needs to trade the training performance for the exploration of rare experiences.

## 5 Conclusion

This paper proposes LKTD as a new sampling framework for deep RL problems via state-space model reformulation. LKTD is equivalent to an accelerated preconditioned SGLD algorithm but with a self-dependent data generating process. For both linear and nonlinear function approximations, LKTD is guaranteed to converge to a stationary distribution $p_*(\theta)$ under mild conditions. Our numerical experiments indicate that LKTD is comparable with Adam algorithm in optimal policy search, while outperforming Adam in robustness and optimal policy explorations. This implies a great potential of LKTD in uncertainty quantification.

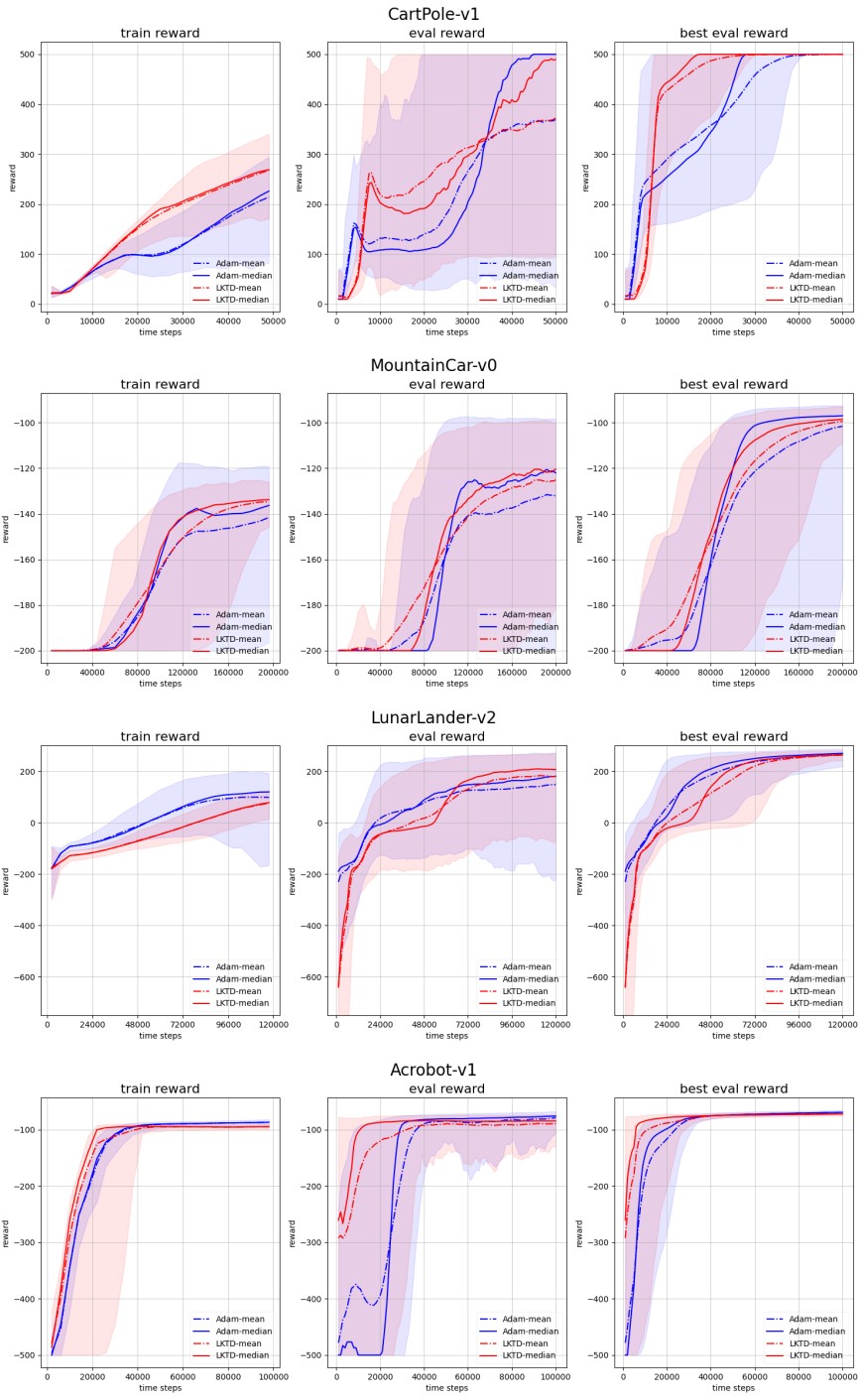

Figure 5: The first column shows the cumulative rewards obtained during the training process, the second column shows the testing performance without random exploration, and the third column shows the performance of best model learnt up to time $t$.

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

## Appendix

## A  COMPLETE PROOFS

### A.1  PROOF OF THEOREM 3.1

PROOF:  Following the proof of LEnKF theorem 3.1 (Zhang et al., 2021), we consider the Kalman gain matrix $K_t = \Omega_t \Phi(\mathbf{x}_t)^\top (R_t + \Phi(\mathbf{x}_t)\Omega_t\Phi(\mathbf{x}_t)^\top)^{-1}$, which can be rewritten as

$$K_t = (I - K_t\Phi(\mathbf{x}_t))\Omega_t\Phi(\mathbf{x}_t)^\top R_t^{-1} = (\Phi(\mathbf{x}_t)^\top R_t^{-1}\Phi(\mathbf{x}_t) + \Omega_t^{-1})^{-1}\Phi(\mathbf{x}_t)^\top R_t^{-1}. \tag{24}$$

Let $\theta_t^f = \theta_{t-1}^a + \delta_t + w_t$, where $\delta_t = \epsilon_t\nabla\log\pi(\theta_{t-1}^a)$. With the identity (24), the conditional expectation of $\theta_t^a$ can be written as

$$
\begin{aligned}
\mathbb{E}(\theta_t^a|\theta_{t-1}^a, \mathbf{r}_t, \mathbf{x}_t) &= \theta_{t-1}^a + \delta_t + K_t(\mathbf{r}_t - \Phi(\mathbf{x}_t)\theta_{t-1}^a - \Phi(\mathbf{x}_t)\delta_t) \\
&= \theta_{t-1}^a + K_t(\mathbf{r}_t - \Phi(\mathbf{x}_t)\theta_{t-1}^a) + (I - K_t\Phi(\mathbf{x}_t))\delta_t \\
&= \theta_{t-1}^a + (I - K_t\Phi(\mathbf{x}_t))\Omega_t\Phi(\mathbf{x}_t)^\top R_t^{-1}(\mathbf{r}_t - \Phi(\mathbf{x}_t)\theta_{t-1}^a) + (I - K_t\Phi(\mathbf{x}_t))\delta_t \\
&= \theta_{t-1}^a + (I - K_t\Phi(\mathbf{x}_t))\Omega_t[\Phi(\mathbf{x}_t)^\top R_t^{-1}(\mathbf{r}_t - \Phi(\mathbf{x}_t)\theta_{t-1}^a) + \Omega_t^{-1}\delta_t] \\
&= \theta_{t-1}^a + \frac{1}{2}(I - K_t\Phi(\mathbf{x}_t))\Omega_t[\Phi(\mathbf{x}_t)^\top V^{-1}(\mathbf{r}_t - \Phi(\mathbf{x}_t)\theta_{t-1}^a) + \Omega_t^{-1}\delta_t] \\
&= \theta_{t-1}^a + \frac{\epsilon_t}{2}\Sigma_t[\Phi(\mathbf{x}_t)^\top V^{-1}(\mathbf{r}_t - \Phi(\mathbf{x}_t)\theta_{t-1}^a) + \nabla\log\pi(\theta_{t-1}^a)],
\end{aligned}
\tag{25}
$$

where $\Sigma_t = I - K_t\Phi(\mathbf{x}_t)$, $\Omega_t = \epsilon_t I$, and $R_t = 2V$. For LKTD, we have

$$
\begin{aligned}
\theta_t^a &= \theta_t^f + K_t(\mathbf{r}_t - \Phi(\mathbf{x}_t)\theta_t^f - v_t) \\
&= \theta_{t-1}^a + \delta_t + w_t + K_t(\mathbf{r}_t - \Phi(\mathbf{x}_t)(\theta_{t-1}^a + \delta_t + w_t) - v_t) \\
&= \mathbb{E}(\theta_t^a|\theta_{t-1}^a, \mathbf{r}_t, \mathbf{x}_t) + w_t - K_t\Phi(\mathbf{x}_t)w_t - K_t v_t \\
&= \mathbb{E}(\theta_t^a|\theta_{t-1}^a, \mathbf{r}_t, \mathbf{x}_t) + e_t,
\end{aligned}
\tag{26}
$$

where $e_t = w_t - K_t(\Phi(\mathbf{x}_t)w_t + v_t)$ with mean $\mathbb{E}(e_t) = 0$ and covariance

$$
\begin{aligned}
Var(e_t) &= Var(w_t) + K_t Var(\Phi(\mathbf{x}_t)w_t + v_t)K_t^\top - 2Cov(w_t, K_t(\Phi(\mathbf{x}_t)w_t + v_t)) \\
&= \Omega_t + K_t(\Phi(\mathbf{x}_t)\Omega_t\Phi(\mathbf{x}_t)^\top + R_t)K_t^\top - 2K_t\Phi(\mathbf{x}_t)\Omega_t \\
&= (I - K_t\Phi(\mathbf{x}_t))Q_t = \epsilon_t\Sigma_t.
\end{aligned}
\tag{27}
$$

By combining (24), (25),(26) and the assumption $V = \sigma^2 I$, the update of $\theta_t^a$ can be rewritten as

$$
\begin{aligned}
\theta_t^a &= \theta_{t-1}^a + \frac{\epsilon_t}{2}\Sigma_t[\Phi(\mathbf{x}_t)^\top V^{-1}(\mathbf{r}_t - \Phi(\mathbf{x}_t)\theta_{t-1}^a) + \nabla\log\pi(\theta_{t-1}^a)] + e_t \\
&= \theta_{t-1}^a + \frac{\epsilon_t}{2}\Sigma_t[\sum_{i=1}^n \frac{1}{\sigma^2}\Phi(x_{t,i})^\top(r_{t,i} - \Phi(x_{t,i})\theta_{t-1}^a) + \nabla\log\pi(\theta_{t-1}^a)] + e_t \\
&= \theta_{t-1}^a + \frac{\epsilon_t}{2}\Sigma_t \sum_{i=1}^n \nabla\log\pi(\theta_{t-1}^a|z_{t,i}) + e_t,
\end{aligned}
\tag{28}
$$

where $\nabla\log\pi(\theta_{t-1}^a|z_{t,i}) = \frac{1}{\sigma^2}\Phi(x_{t,i})^\top(r_{t,i} - \Phi(x_{t,i})\theta_{t-1}^a) + \frac{1}{n}\nabla\log\pi(\theta_{t-1}^a)$. $\square$

### A.2  LEMMA FOR THEOREM 3.2

We assume the following conditions hold:

(A1) For any $\theta \in \Theta$, the Markov transition kernel $\Pi_\theta$ has a single stationary distribution $\pi_\theta(z)$, $G : \Theta \times \mathcal{Z}$ is measurable, and $\|g(\theta)\| = \|\int_{\mathcal{Z}} G(\theta, z)\pi(z|\theta)dz\| < \infty$.

(A2) There exists a function $\mathcal{G}(\theta, z)$, which is an anti-derivative of $G(\theta, z)$ with respect to $\theta$, i.e., $\nabla_\theta\mathcal{G}(\theta, z) = G(\theta, z)$, such that $|\mathcal{G}(0, z)| \leq A$ for some constant $A > 0$ and any $z \in \mathcal{Z}$; in addition, there exists some constant $B > 0$ such that $\|G(0, z)\| \leq B$ for any $z \in \mathcal{Z}$.

(A3) There exists some constant $M > 0$ such that for any $z \in \mathcal{Z}$,

$$\|G(\theta, z) - G(\vartheta, z)\| \le M\|\theta - \vartheta\|, \quad \forall \theta, \vartheta \in \Theta.$$

(A4) For each $z \in \mathcal{Z}$, the function $G(\cdot, z)$ is $(m, b)$-dissipative; for some $m > 0$ and $b \ge 0$,

$$\langle \theta, G(\theta, z) \rangle \ge m\|\theta\|^2 - b, \quad \forall \theta \in \Theta.$$

(A5) There exist a constant $\delta \in [0, 1)$ and some constants $M$ and $B$ such that

$$\mathbb{E}\|G(\theta, z) - g(\theta)\|^2 \le 2\delta(M^2\|\theta\|^2 + B^2), \quad \forall \theta \in \Theta.$$

(A6) The probability law $\mu_0$ of the initial hypothesis $\theta_0$ has a bounded and strictly positive density $p_0$ with respect to the Lebesgue measure on $\Theta$, and

$$\kappa_0 := \log \int_\Theta e^{\|\theta\|^2} p_0(\theta) d\theta < \infty.$$

**Lemma A.1** *(Proposition 10 of Raginsky et al. (2017)) Consider the SGLD algorithm with a constant learning rate $\epsilon$,*

$$\theta_t = \theta_{t-1} - \epsilon G(\theta_{t-1}, z_t) + \sqrt{2\epsilon\beta^{-1}} e_t, \tag{29}$$

*where $e_t \sim N(0, I_d)$, $d$ is the dimension of $\theta$, and $\beta$ is the inverse temperature. Assume the conditions (A1)-(A6) hold. If $\mathbb{E}G(\theta_{t-1}, z_t) = g(\theta_{t-1})$ holds for any step $t \in \mathbb{N}$, $\beta \ge 1 \vee \frac{2}{m}$, and $0 < \epsilon < 1 \wedge \frac{m}{4M^2}$, then*

$$W_2(p_t, p_*) \le (\tilde{C}_0 \delta^{1/4} + \tilde{C}_1 \epsilon^{1/4}) t\epsilon + \tilde{C}_2 e^{-t\epsilon/\beta C_{LS}}, \tag{30}$$

*where $p_t(\theta)$ denotes the density function of $\theta_t$; $p_*(\theta) \propto \exp(-\beta\tilde{\mathcal{G}}(\theta))$, $\tilde{\mathcal{G}}(\theta)$ is the anti-derivative of $g(\theta)$, i.e., $\nabla_\theta \tilde{\mathcal{G}}(\theta) = g(\theta)$; $C_{LS}$ denotes a logarithmic Sobolev constant satisfied by the $p_*$, and the constants $\tilde{C}_0$, $\tilde{C}_1$ and $\tilde{C}_2$ are given by*

$$C_0 = \left( M^2 \left( \kappa_0 + 2 \left( 1 \vee \frac{1}{m} \right) \left( b + 2B^2 + \frac{d}{\beta} \right) \right) + B^2 \right),$$

$$C_1 = 6M^2(\beta C_0 + d),$$

$$\tilde{C}_0 = \sqrt{(12 + 8(\kappa_0 + 2b + \frac{2d}{\beta}))(\beta C_0 + \sqrt{\beta C_0})},$$

$$\tilde{C}_1 = \sqrt{(12 + 8(\kappa_0 + 2b + \frac{2d}{\beta}))(C_0 + \sqrt{C_0})},$$

$$\tilde{C}_2 = \sqrt{2C_{LS} \left( \log\|p_0\|_\infty + \frac{d}{2}\log\frac{3\pi}{m\beta} + \beta \left( \frac{M\kappa_0}{3} + B\sqrt{\kappa_0} + A + \frac{b}{2}\log 3 \right) \right)}.$$

PROOF: The proof of Lemma A.1 follows from Proposition 10 of Raginsky et al. (2017). □

## A.3 PROOF OF THEOREM 3.2

PROOF: By Theorem 1 of Ma et al. (2015), Algorithm (16) (of the main text) works as a pre-conditioned SGLD algorithm with the pre-conditioner $\Sigma_t$, and it has the same stationary distribution as the algorithm (29). By (30), we have $W_2(p_T, p_*) \to 0$ for algorithm (29) under the given settings of $\epsilon$ and $N_T$. Therefore, for Algorithm (16), we also have $W_2(p_T, p_*) \to 0$ as $T \to \infty$ by noting that $\Sigma_t$ is positive definite for any $t$. □

## A.4 PROOF OF LEMMA 3.1

PROOF: Since we assume that all samples are *i.i.d*, it suffices to prove the lemma with the case $n = 1$. In this case, $G(\theta, \mathbf{z}) = G(\theta, z) = -\frac{1}{\sigma^2}\Phi(x)^\top(r - \Phi(x)\theta) + \frac{1}{\sigma_\theta^2}\theta$. Let $g(\theta) = \mathbb{E}_{z \sim \mu_\theta}[G(\theta, z)] = \int_\mathcal{Z} G(\theta, z)\pi(z|\theta)dz$ be the expected gradient with respect to the stationary distribution $\mu_\theta$.

(A1) By assumption *(i)* and *(ii)*, then by simple algebra

$$\|G(\theta, z)\| \le \frac{1}{\sigma^2}\|\Phi(x)\| \cdot \|(r - \Phi(x)\theta)\| + \frac{1}{\sigma_\theta^2}\|\theta\|$$

$$\le \frac{1}{\sigma^2}(1 + \gamma)(\bar{r} + (1 + \gamma)\|\theta\|) + \frac{1}{\sigma_\theta^2}\|\theta\| \tag{31}$$

$$= \frac{1}{\sigma^2}(1 + \gamma)\bar{r} + (\frac{1}{\sigma^2}(1 + \gamma)^2 + \frac{1}{\sigma_\theta^2})\|\theta\| < \infty.$$

Since the upper bound is independent of $z$, the expected gradient $g(\theta)$ is well-defined with $\|g(\theta)\| < \infty$ for all $\theta \in \Theta$.

(A2) Given the explicit formulation of $G(\theta, z)$, the anti-derivative $\mathcal{G}(\theta, z)$ can be derived as

$$\mathcal{G}(\theta, z) = \frac{1}{2\sigma^2}(r - \Phi(x)\theta)^2 + \frac{1}{2\sigma_\theta^2}\|\theta\|^2. \tag{32}$$

For any $z \in \mathcal{Z}$, we can derive the following bound

$$|\mathcal{G}(0, z)| = \frac{1}{2\sigma^2}r^2 \le \frac{1}{2\sigma^2}\bar{r}^2, \tag{33}$$

and

$$\|G(0, z)\| \le \frac{1}{\sigma^2}(1 + \gamma)\bar{r}. \tag{34}$$

(A3) For any $z \in \mathcal{Z}$,

$$\|G(\theta, z) - G(\vartheta, z)\| = \|\frac{1}{\sigma^2}\Phi(x)^\top\Phi(x)(\theta - \vartheta) + \frac{1}{\sigma_\theta^2}(\theta - \vartheta)\|$$

$$\le (\frac{1}{\sigma^2}\|\Phi(x)\|^2 + \frac{1}{\sigma_\theta^2})\|\theta - \vartheta\| \tag{35}$$

$$= (\frac{1}{\sigma^2}(1 + \gamma)^2 + \frac{1}{\sigma_\theta^2})\|\theta - \vartheta\|.$$

(A4) For any $z \in \mathcal{Z}$,

$$\langle\theta, G(\theta, z)\rangle = -\frac{1}{\sigma^2}(\Phi(x)\theta)(r - \Phi(x)\theta) + \frac{1}{\sigma_\theta^2}\|\theta\|^2$$

$$= \frac{1}{\sigma^2}(\Phi(x)\theta - \frac{r}{2})^2 - \frac{r^2}{4\sigma^2} + \frac{1}{\sigma_\theta^2}\|\theta\|^2 \tag{36}$$

$$\ge \frac{1}{\sigma_\theta^2}\|\theta\|^2 - \frac{\bar{r}^2}{4\sigma^2}.$$

(A5) Since $\|G(\theta, z)\|$ is uniformly bounded for all $z \in \mathcal{Z}$ given in (31), the gradient bias is bounded by

$$\|G(\theta, z) - g(\theta)\| \le 2\max_{z \in \mathcal{Z}}\|G(\theta, z)\| \le \frac{2}{\sigma^2}(1 + \gamma)\bar{r} + 2(\frac{1}{\sigma^2}(1 + \gamma)^2 + \frac{1}{\sigma_\theta^2})\|\theta\| \tag{37}$$

By some algebra, we can calculate the quadratic bound

$$\|G(\theta, z) - g(\theta)\|^2 \le \delta(M^2\|\theta\|^2 + B^2), \tag{38}$$

where $M^2 = 4(\frac{1}{\sigma^2}(1 + \gamma)^2 + \frac{1}{\sigma_\theta^2})^2 + 1$ and $B^2 = M^2(\frac{2}{\sigma^2}(1 + \gamma)\bar{r})^2$. Since the bound is uniform for all $z \in \mathcal{Z}$, we can derive the desired bound for the expectation.

(A6) By assumption *(iii)*,

$$\kappa_0 = \log\int_\Theta e^{\|\theta\|^2}e^{\frac{-1}{2\sigma_0^2}\|\theta\|^2}d\theta - p\log\sqrt{2\pi\sigma_0^2}$$

$$< \log\int_\Theta e^{(1 - \frac{1}{2\sigma_0^2})\|\theta\|^2}d\theta < \infty, \tag{39}$$

for any $\sigma_0^2 < \frac{1}{2}$.

□

## B  MORE NUMERICAL RESULTS

### B.1  SOFTMAX PROBABILISTIC POLICY

For the indoor escape environment, we adopt the Boltzmann exploration which selects an action $a$ with probability

$$P_{\rho_\theta}(a|s) = \frac{\exp\{\beta_{\text{act}}Q(s,a;\theta)\}}{\sum_{a'\in\mathcal{A}}\exp\{\beta_{\text{act}}Q(s,a';\theta)\}},$$

where $\beta_{\text{act}}$ is the action inverse temperature. When $\beta_{\text{act}}$ is small, the agent tends to explore random actions. In contrast, when $\beta_{\text{act}}$ is large, the agent takes action greedily. Greedy Q-learning can be viewed as a special case of Boltzmann exploration, since $\rho_\theta(s) = \text{argmax}_a Q(s,a;\theta)$ with probability 1 as $\beta_{\text{act}} \to \infty$. Moreover, the action probability $P(\rho_\theta(s) = a)$ is differentiable with respect to $\theta$. In this paper, we assume all RL algorithms follow the $\epsilon$-Boltzmann exploration with the action probability given by

$$P_{\rho_\theta^\epsilon}(a|s) = \epsilon + (1-\epsilon)\frac{\exp\{\beta_{\text{act}}Q(s,a;\theta)\}}{\sum_{a'\in\mathcal{A}}\exp\{\beta_{\text{act}}Q(s,a';\theta)\}}, \tag{40}$$

where $\epsilon$ is the random exploration rate. Note that as $\beta_{\text{act}} \to \infty$, $\epsilon$-Boltzmann exploration converges to $\epsilon$-greedy exploration.

### B.2  STATE VALUE VISUALIZATION FOR THE INDOOR ESCAPING EXAMPLE

In this section, we demonstrate the state value approximation for both linear and nonlinear function approximations. By following Melo & Ribeiro (2007), we define the Q-function and the feature map as

$$Q(s,a) = \phi(s)^\top\theta = \sum_{a'\in\mathcal{A}}\bar{\phi}(s)^\top\theta_{a'}\mathbf{1}_{a'}(a) \quad\text{and}\quad \bar{\phi}(s) = (\sigma_1(x,y), \sigma_2(x,y), \ldots, \sigma_4(x,y))^\top, \tag{41}$$

where the parameter vector $\theta = (\theta_N^\top, \theta_S^\top, \theta_E^\top, \theta_W^\top)^\top$ is partitioned into 4 subspaces, $\mathbf{1}(\cdot)$ is the indicator function, and $\sigma_i : \mathcal{S} \to \mathbb{R}$ is a basis function which can be a linear basis function, Gaussian kernel, etc. For each set of basis functions, the agent is allowed to update its parameter every 60 steps with 20 transition tuples as training data for 30,000 episodes. In every experiment, we collect the last 1,000 updates as the samples from the stationary distribution. In figure 6, we compare different function approximation of SARSA type LKTD. The optimal policy used to simulate the true state-value is defined as

$$\rho^*(x,y) = \begin{cases} N, & \text{if } x \geq y, \\ E, & \text{if } x < y, \end{cases} \tag{42}$$

where the discount factor $\gamma = 0.9$. In figure 6b, 6c and 6d, the state-value surfaces are estimated using the average state-values of last 1000 updates with respect to linear-based approximation, kernel-based approximation and deep neural network approximation respectively. We showed that LKTD algorithm successfully estimate the state-values for all three function approximations.

### B.3  CONTINUATION OF UNCERTAINTY QUANTIFICATION FOR THE INDOOR ESCAPING EXAMPLE

This section is a supplement to Section 4.1 of the main text, which includes more numerical results for comparison of the proposed LKTD algorithm and the popular Adam algorithm (Kingma & Ba, 2014).

In each run of LKTD, we set the batch size to 250, fix the inverse temperature $\beta = 1$, and update the network parameters every 50 steps. For $\sigma$ and $\sigma_\theta$ in Remark 1, we set $\sigma^2 = 1$ and $\sigma_\theta^2 = 25$, where $\sigma^2$ is estimated by the mean square TD error according to Assumption 2, and $\sigma_\theta^2$ is chosen suitably for convergence and ignorable prior effects. Under the setting $\beta = 1$, LKTD converges to the stationary distribution $p_*(\theta) \propto \exp(-\tilde{\mathcal{G}}(\theta))$. The parameters obtained in the last 1000 parameter updates are used as a parameter ensemble for performing the followed Bayesian inference tasks. The parameter ensemble naturally induces a Q-value ensemble and a policy ensemble. More precisely, each parameter vector corresponds to a Q-function and a greedy policy induced from the Q-function.

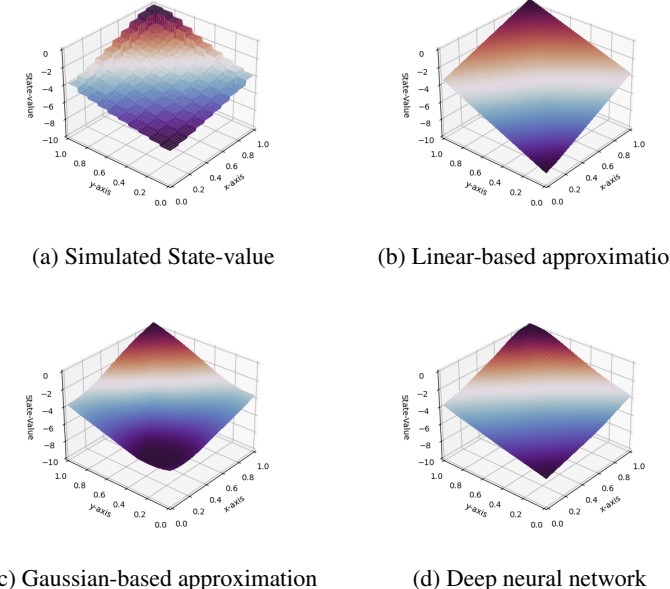

(a) Simulated State-value        (b) Linear-based approximation

(c) Gaussian-based approximation        (d) Deep neural network

Figure 6: State-value surface: (a) State-values of the center point in each grid with respect to the optimal policy. (b) Linear function approximation with linear basis. (c) Linear function approximation with Gaussian kernel. (d) Deep neural network with hidden-layers (16,16,16).

The experimental results are reported in Tables 2, 3 and 5, where each measurement is derived by averaging over 200 independent runs.

For comparison, Adam has been run with different learning rates, including 1.0e-2, 1.0e-3 and 1.0e-4. For each learning rate, it is run for 200 times independently, each run consisting of 10000 episodes. Similar to LKTD, we use the parameters obtained in the last 1000 parameter updating steps as a parameter ensemble for performing the followed statistical inference tasks. The results are also summarized in Tables 2, 3 and 5.

Table 2 reports the estimation accuracy of Q-values, which is measured by the mean squared error between the mean of the Q-value ensemble and the Q-value of the optimal policy as defined in equation 42. It is easy to see that Adam produces about the same MSEs for all four actions with a learning rate of 1e-2, and it produces more varied MSEs with other learning rates. LKTD produces almost the same MSEs as the best run of Adam.

Table 2: MSE($\hat{Q}$) for the indoor escaping example

| Description | | North | East | South | West | Average |
|---|---|---|---|---|---|---|
| Name | Learning rate | MSE($\hat{Q}$) | MSE($\hat{Q}$) | MSE($\hat{Q}$) | MSE($\hat{Q}$) | MSE($\hat{Q}$) |
| **LKTD** | **1e-4** | **0.119** | **0.120** | **0.100** | **0.101** | **0.110** |
| **Adam** | **1e-2** | **0.113** | **0.113** | **0.096** | **0.096** | **0.105** |
| Adam | 1e-3. | 0.130 | 0.139 | 0.354 | 0.355 | 0.245 |
| Adam | 1e-4 | 0.120 | 0.119. | 0.396 | 0.392 | 0.257 |

Table3 compares the performance of the two algorithms in optimal policy exploration, which is measured by MSE($\hat{p}$), the mean squared error between the proportions of action votes from the policy ensemble and the probabilities of mean optimal actions. The probabilities of mean optimal actions describe the variety of optimal actions at a state. That is, if multiple actions are all optimal, the probabilities of mean optimal actions are the same across all optimal actions. For example, suppose that both action N and action E are optimal at a state, then each has a mean optimal action probability

of 0.5; therefore, a policy ensemble that fails to explore all optimal policies (due to a local trap issue) might only vote for one of the two actions. For LKTD, the smaller values of MSE($\hat{p}$) in the north and east actions imply that it provides better optimal policy exploration than Adam.

It is worth mentioning that Adam with a learning rate of 1e-2 also produces similar MSE($\hat{p}$) values to LKTD, but its SOAR in table 4 is worse than LKTD, which implies that LKTD provides a more reliable policy than Adam.

Table 3: MSE($\hat{p}$) for the indoor escaping example

| Description | | North | East | South | West | Average |
|---|---|---|---|---|---|---|
| Name | Learning rate | MSE($\hat{p}$) | MSE($\hat{p}$) | MSE($\hat{p}$) | MSE($\hat{p}$) | MSE($\hat{p}$) |
| LKTD | 1e-4 | **0.044** | **0.044** | 0.002 | 0.002 | 0.023 |
| Adam | 1e-2 | 0.044 | 0.044 | 0 | 0 | 0.022 |
| Adam | 1e-3 | 0.047 | 0.047 | 0 | 0 | 0.0235 |
| Adam | 1e-4 | 0.058 | 0.058 | 0 | 0 | 0.029 |

Table 4: Sub-optimal action rate for the indoor escaping example

| Description | | North | East | South | West | |
|---|---|---|---|---|---|---|
| Name | Learning rate | SOAR | SOAR | SOAR | SOAR | Average |
| LKTD | 1e-4 | **0.098** | **0.102** | **0.026** | **0.026** | **0.063** |
| Adam | 1e-2 | 0.134 | 0.135 | 0.005 | 0.005 | 0.070 |
| Adam | 1e-3 | 0.493 | 0.495 | 0 | 0 | 0.247 |
| Adam | 1e-4 | 0.502 | 0.495 | 0 | 0 | 0.248 |

Table 5 compares the coverage rates of the optimal Q-values by different algorithms. By considering 100 grid points over the entire state space, we can calculate the coverage rate of optimal Q-values. Table 5 shows that LKTD has consistent coverage rates around 95% for all actions, while Adam with small learning rates failed to cover over 50% of the optimal Q-values. Although Adam with a large learning rate (1e-2) can provide a good exploration for optimal policies, however, due to its optimization nature, it cannot provide a correct confidence coverage for the Q-value.

Table 5: Coverage rate of the optimal Q-value for the indoor escaping example

| Description | | North | East | South | West | |
|---|---|---|---|---|---|---|
| Name | Learning rate | CR | CR | CR | CR | Average |
| LKTD | 1e-4 | **0.935** | **0.928** | **0.972** | **0.968** | **0.951** |
| Adam | 1e-2 | 0.883 | 0.883 | 0.887 | 0.884 | 0.884 |
| Adam | 1e-3 | 0.635 | 0.634 | 0.314 | 0.316 | 0.475 |
| Adam | 1e-4 | 0.263 | 0.263 | 0.109 | 0.108 | 0.186 |

**Remark 2** *Instead of using semi-gradient, the true gradient is used in all of the indoor escaping experiments of LTKD and Adam. Note that the semi-gradient is biased, which can lead to incorrect stationary distribution. In order to keep the comparison subjective, the objective function for Adam at each time $t$ is given by*

$$L(\theta) = \frac{1}{n} \sum_{i=1}^{n} (Q(s_i, a_i; \theta) - r_i - \gamma Q(s_i', a_i'; \theta))^2. \tag{43}$$

## B.4 HYPERPARAMETER SETTINGS FOR CLASSIC CONTROL PROBLEMS

Our experiment is based on the framework of RL Baselines3 Zoo. For Adam optimizer, the hyperparameters are provided by Zoo package. For LKTD, we set $\sigma = 1$ and $1/\beta = 0.01$, and $\sigma_\theta$ can be

chosen suitably according to the parameter size. The DQN agents are trained using a 2-layer dense neural network with hidden layers of size (256, 256). All the hyperparameters are shown in table 6. Note that if the gradient step is set to -1, the agent conducts as many gradient steps as steps done in the environment between two updates.

Table 6: Hyperparameters

| Environment | CartPole-v1 | | MountainCar-v0 | | LunarLander-v2 | | Acrobot-v1 | |
|---|---|---|---|---|---|---|---|---|
| Hyperparameters | LKTD | Adam | LKTD | Adam | LKTD | Adam | LKTD | Adam |
| learning rate | 2.5e-5 | 2.3e-3 | 1e-4 | 4e-3 | 5e-6 | 6.3e-4 | 5e-5 | 6.3e-4 |
| $1/\beta$ (temperature) | 0.01 | - | 0.01 | - | 0.01 | - | 0.01 | - |
| $\sigma_\theta$ (prior) | **5** | - | **5** | - | **20** | - | **5** | - |
| $\sigma$ (observation) | 1 | - | 1 | - | 1 | - | 1 | - |
| target update interval | 1 | 1000 | 1 | 600 | 1 | 250 | 1 | 250 |
| $\gamma$(discount factor) | 0.99 | | 0.98 | | 0.99 | | 0.99 | |
| training steps | 5e4 | | 2e5 | | 1.2e5 | | 1e5 | |
| batch size | 64 | | 64 | | 64 | | 64 | |
| learning starts | 1e5 | | 1e3 | | 0 | | 0 | |
| train freq | 256 | | 16 | | 4 | | 4 | |
| gradient steps | 128 | | 8 | | -1 | | -1 | |
| exploration fraction | 0.16 | | 0.2 | | 0.12 | | 0.12 | |
| exploration final eps | 0.04 | | 0.07 | | **0.25** | 0.1 | 0.1 | |

