# OpenReview forum: "A sampling framework for value-based reinforcement learning"
_ICLR.cc/2023/Conference — Submitted to ICLR 2023_

### Official Review · Reviewer_2cQs · 2022-10-24

**Confidence:** 3
**Clarity, Quality, Novelty And Reproducibility:** See weaknesses above
**Correctness:** 2
**Technical Novelty And Significance:** 2
**Empirical Novelty And Significance:** 1
**Recommendation:** 1

**Strength And Weaknesses:**

There are a few things to like about the paper:
- This paper is taking on a "big" problem that would be of very high value to the ML community.
- The general idea of using Langevin-style updates is appealing, and may hold promise in the future.
- The general progression of algorithms, theorems and then empirical results is a good structure.

However, there are some significant shortcomings in this paper:
- There are many pieces of this algorithm that don't quite make sense... why go through all the problems of Bayesian inference to then use "epsilon-Boltzmann exploration"? This is clearly an inefficient dithering scheme, if you were going to use this then you really give up the main lure of Bayesian RL.
- There are some important holes in the related literature, to highlight two:
  1. Deep Exploration via Randomized Value Functions (Osband et al)
  2. Langevin DQN (Dwaracherla + Van Roy)
  - reviewing these papers + related works in the area will be very important to help make sense of this work. Paper (1) gives an overview of the importance of "deep epxloration" and possible ways to have tractable posterior approximation in deep RL. Paper (2) explores an approach very linked to the LKTD that you propose.
- The quality/clarity of the experimental results is lacking... none of the curves are separated beyond the confidence intervals + the the effects are confusing overall. I would suggest using something like the "DeepSea" environments available in opensource "bsuite" to give you something clear and simple to get started with... similar to paper (2)

**Summary Of The Paper:**

This paper proposes a Langevin-style approach to posterior approximation in TD learning.
The idea is that using this style of MCMC learning we can get useful uncertainty estimates, and better decision making as we scale to deep RL.

**Summary Of The Review:**

Overall I think this paper is taking on an interesting problem, with a potentially interesting approach.
However, there are some glaring holes in the related work... also, the experimental results are a bit muddled/unclear.
I think that, at a minimum, revisiting the paper in light of that related work should be the first step for this paper.
In its current form, I do not recommend acceptance.

---

### Official Review · Reviewer_soqG · 2022-10-24

**Confidence:** 2
**Correctness:** 2
**Technical Novelty And Significance:** 2
**Empirical Novelty And Significance:** 2
**Recommendation:** 3

**Clarity, Quality, Novelty And Reproducibility:**

I believe the paper has serious issues with clarity that make it hard to review which I indicated above.

Some other nits I noted down while reading:

* In section 2.2 above Eq (1) has $\theta \in \theta$

* Using \rho for policy and \pi for prior might make the paper a bit less readable compared to following the usual notation of \pi for policy.

* Eq (2) has n in the expectation without definition.

* The notation for $\tilde{\mathcal{F}}_z(\theta)$ is inconsistent with a tilde missing sometimes (e.g. in first line of pg3), and the $\theta$ dependence missing in some other places (eg. Eq (3)).

* Add a citation for "it has been proven under some regularity assumption"... in L3 of pg 3.

* Eq (5) looks odd, since its not a real conditional switch -- might be best to use one and say that the action ablated version is straightforward to generalize to? Same thing about simplifying Eqs (6) and (9) to use one particular setting since they are both identical.

* What is the $\pi$ in Eq (11)? Is it the same as $\pi_*$?

* Algorithm 1 lacks a description of $r^f_t$.


**Details Of Ethics Concerns:**

No particular concerns

**Strength And Weaknesses:**

The most serious issue in my opinion is clarity and the assumptions made wrt theory that seem hard to justify. For example, Assumption 2 appears to be completely unreasonable -- $\theta_t$ is the parameter for the model, and the assumption says that the environment changes at each time step to fit the current model.

In connection with dynamic policy, what is the precise definition of the map that induces a stationary distribution for a given $\theta$? This is never specified.

There is also a serious problem with clarity, where the density function in Assumption is called $\pi(z_t | \theta_{t-1})$. My understanding is that the authors use $\pi$ for a prior on the parameter space $\theta$. How did this prior in the param space morph into a distribution on the samples?





**Summary Of The Paper:**

This paper presents an approach to quantify the uncertainty in value based RL algorithms such as TD learning and Q-learning.  The approach is based on prior work that uses a state-space modeling approach to derive a Kalman filtering algorithm for TD learning (KTD and KOVA). Unless the function approximation is linear and a normality assumption holds, this requires approximations that have a quadratic complexity in the number of parameters.  Instead, the current work has the goal of applying Langevinized ensemble approach to the value estimation setting, in order to get a lower complexity.

**Summary Of The Review:**

I believe the assumptions are completely unreasonable as I mentioned above. Besides that, there are serious issues with the clarity and significance due to the incremental nature of applying prior work. The main delta compared to previous work appears to be in applying the prior work to the linear model defined by the TD loss, which seems minor even ignoring the assumptions and clarity.

---

### Official Review · Reviewer_ZkBy · 2022-10-25

**Confidence:** 3
**Correctness:** 3
**Technical Novelty And Significance:** 2
**Empirical Novelty And Significance:** 2
**Recommendation:** 3

**Clarity, Quality, Novelty And Reproducibility:**

As stated above, the novelty and significance of this work seem minor to me based on the current paper. At least I would like to see experiments that: (1) show LKTD is efficient compared with e.g. KTD, KOVA in terms of dimension dependence, hence validating the novelty stated at the end of section 1; (2) compare LKTD with KTD, KOVA in terms of MSE, SOAR to validate a non-trivial improvement in robustness and exploration.

Minor questions/typos:
- What's the definition of $W_2(\cdot,\cdot)$?
- What's Figure 4b showing? Is 'figure 3b' in the last sentence on page 7 a typo?

**Strength And Weaknesses:**

## Strengths
- This paper is in general well-written and relatively easy to follow. The proof seems to be theoretically sound.
## Weakness
I worry about the novelty and significance of this work.

- For instance, it is stated in the major contributions that the new algorithm is scalable to dimension, and is computationally efficient, but I don't see a clear comparison of this work with KTD, KOVA in terms of computational complexity.
- Also, all experiments are comparing LKTD with Adam rather than KTD, KOVA. I wonder is there a significant improvement compared with these two methods.

**Summary Of The Paper:**

 This paper proposed a sampling framework LKTD for value-based RL that converges to a stationary distribution and enables efficient sample generation from the posterior. They show numerical experiments that indicate LKTD is comparable to Adam while outperforming it in robustness.

**Summary Of The Review:**

In all, although this paper is well-written and seems to be theoretically sound, I believe for now its novelty and significance (especially over KTD, KOVA) haven't been validated by its experiments. I would like to see more experiments focusing on these aspects in the revision if possible.

---

### Official Review · Reviewer_Je2E · 2022-10-29

**Confidence:** 4
**Correctness:** 3
**Technical Novelty And Significance:** 2
**Empirical Novelty And Significance:** 2
**Recommendation:** 3

**Clarity, Quality, Novelty And Reproducibility:**

The paper is well organized, and I think most of the writings are clear. However, I think some terminologies are kind of confusing when I first read the paper. For example, the title is "A sampling framework", but in fact the paper is about a new algorithm to learn parameters of a value function. The paper says "a state $\theta$", but it's actually the parameter to be learned. My conjecture is that the authors are from another field. I am not saying that using these new terminologies is wrong, and it is always great to see researchers from other fields also care about RL and machine learning in general. However, I think maybe it's better to make some efforts to bridge the "cultural gap", so that the paper is more readable for the potential audience.


**Strength And Weaknesses:**

Strength

The paper provides a novel perspective to study value-based RL. The main idea is to optimize MSBE using SGLD which is novel to me. Following this perspective, convergence guarantee of the proposed algorithm can be derived. The paper also provides empirical study to show the effectiveness of the algorithm.

Weakness

As discussed above, in my opinion the main contribution of the paper is to use SGLD to optimize MSBE. So the key to decide if the contribution is significant enough is to understand if this new method can introduce some advantage from either empirical or theoretical perspective. However this is not clear to me as explained in details below. Please correct me if I missed anything important. I am happy to adjust my score based on how well the authors answer the questions in the rebuttal.

1. One classical method in RL is the residual gradient (RG), which directly optimizes MSBE using gradient descent [1]. In fact, as discussed in Remark 1, the gradient is the same as RG (plus a regularization of the parameter). As a gradient-based method, the convergence of RG is robust with both linear and non-linear function approximation. I am almost sure that under the data collection assumption used in the paper, it can also be proved that RG converges. If that’s the case, what’s the advantage of LKTD compared to RG?

2. It is known that one problem of RG is the double sampling issue. That is, in stochastic environment, to obtain an unbiased gradient estimate, we need two independent samples. It seems to me that LKTD has the same problem?

3. The author suggests that equation 23 can be used to connect LKTD with DQN. However, besides the semi-gradient TD update, DQN also has other components such as replay buffer and target network. Also, the data stored in the replay buffer is collected from a very different way from Assumption 1. How do these fit the analysis of LKTD?

4. It seems that one of the main contributions is to prove LKTD converges under Assumption 1. However, Assumption 1 basically considers an on-policy setting, in which case the convergence of TD with function approximation can also be proved, see [2,3] for example. Again, what’s the advantage of LKTD compared to TD?

5. There are no related works about RG and TD discussed in the paper.

6. It’s good to see that the paper also test LKTD empirically. But from the experiment results, it is now clear if LKTD significantly outperforms a TD based algorithm. Also, the test domains might be too simple. I think it will be very convincing if the author can show competitive performance compared to TD based algorithm in more challenging  problems.

7. In the experiments the baseline is Adam. But Adam is an optimization algorithm, what's the objective used in the baseline algorithm? For the control problem, does it mean DQN with Adam as the optimizer?

[1] Baird, L., 1995. Residual algorithms: Reinforcement learning with function approximation. In Machine Learning Proceedings 1995 (pp. 30-37). Morgan Kaufmann.

[2] Bhandari, J., Russo, D. and Singal, R., 2018, July. A finite time analysis of temporal difference learning with linear function approximation. In Conference on learning theory (pp. 1691-1692). PMLR.

[3] Tsitsiklis, J. and Van Roy, B., 1996. Analysis of temporal-diffference learning with function approximation. Advances in neural information processing systems, 9.


**Summary Of The Paper:**

This paper proposes Langivinized Kalman Temporal Difference (LKTD) learning, a new algorithm for value based reinforcement learning (RL). LKTD optimizes the mean squared Bellman error (MSBE) objective function using SGLD, in contrast to classical algorithm such as residual gradient which applies gradient descent, and temporal difference learning which uses semi-gradient update. The paper proves the convergence of LKTD and provides empirical studies in synthetic problems and simple control problems.

**Summary Of The Review:**

I recommend rejecting this paper as the advantage of the proposed algorithm is not clear at this point. I hope the authors can address my questions in the rebuttal.

---

### Decision · Program_Chairs · 2023-01-20

**Decision:**

Reject

**Justification For Why Not Higher Score:**

To reiterate, all reviewers argued that he paper was a clear rejection, with scores ranging from 1-3. The authors decided to comment on the reviewers concerns. I believe this is such a clear case that a very short meta review suffices.

**Justification For Why Not Lower Score:**

N/A

**Metareview: Summary, Strengths And Weaknesses:**

The paper uses a Langevin MCMC style approach together with TD learning to form 'posterior' distributions over value functions. The paper studies an interesting problem, but reviewers point out severe issues with evaluation, technical assumptions, and novelty with respect to uncited past work.

All reviewers argued that he paper was a clear rejection, with scores ranging from 1-3. The authors decided to comment on the reviewers concerns. I believe this is such a clear case that a very short meta review suffices.